# Demand–supply-side barriers affecting maternal health service utilization among rural women of West Shoa Zone, Oromia, Ethiopia: A qualitative study

**Seifadin Ahmed Shallo**[1]*, **Deresa Bekele Daba**[2]°, **Abuzumeran Abubekar**[2]°

**1** Department of Public Health, College of Medicine and Health Sciences, Ambo University, Ambo, Ethiopia,
**2** Division of Epidemiology, Department of Public Health, Ambo University, Ambo, Ethiopia

° These authors contributed equally to this work.
* Seifadinahmed8226@gmail.com

## Abstract

### Introduction

Despite the efforts and strategies being applied by the government and the partner organizations to increase maternal health service utilization, maternal health service utilization is low in the general population and very low in rural communities of the West Shoa Zone specifically.

### Objective

This study intended to identify and describe barriers contributing to low maternal health service utilization in selected rural districts of the West Shoa Zone of Oromia regional state, Ethiopia, by 2021.

### Methods

The study was conducted from February 01 to April 30/2021 in three districts of the West Shoa Zone. The districts were selected purposively based on the report of their last year's (2020) performance on maternal health service utilization obtained from the Zonal health office, where the ANC follow-up and Institutional delivery were the lowest among the Districts in the Zone. A community-based qualitative phenomenological approach was used to explore the demand-supply side barriers affecting the utilization of maternal health services. Six Focus Group discussions, 9 In-depth Interviews, and 12 Key Informants Interviews were conducted with women who gave birth at home in the last 12 months and with health care providers at different health offices and health institutions. Data were tape-recorded, transcribed verbatim, translated, and analyzed thematically using MAXQDA software.

### Results

Our findings revealed that though women strongly agree on the importance and advantage of maternal health services utilization, both demand and supply side barriers such as low

**Data Availability Statement:** All relevant data are within the paper and its Supporting information files.

**Funding:** The author(s) received no specific funding for this work.

**Competing interests:** The authors have declared that no competing interests exist.

**Abbreviations:** ANC, Antenatal care; CRC, compassionate, respectful, and care; EDD, Estimated Date of Delivery; FGD, Focus Grip discussion; H/C, Health center; HDA, Health development army; HEW, Health Extension Worker; ID, Institutional delivery; IDI, In-depth interview; IMR, Infant Mortality Rate; KII, key Informant interview; MCH, Maternal and Child Health; MM, maternal mortality ratio; SDG, sustainable development goal.

awareness on when to use the services, not knowing ANC schedule, misinformation about ANC and institutional delivery, not knowing their estimated date of delivery and precipitated labor, shortage of manpower at health institutions, far distance health facilities, and unavailability or un-accessibility of ambulance services during an emergency time, lack of transportation to health facilities were hindering them not to use the services.

## Conclusion

In general, rural women are facing many challenges yet to accessing and utilizing maternal health services. To achieve the SDG targets, addressing barriers prohibiting a woman from using MCH should be critically addressed.

## I. Introduction

Maternal health refers to the health of the women during pregnancy, childbirth, and the postpartum period. Maternal health services are services that a mother should access so that she will undergo healthy and safe pregnancy, delivery, and the postpartum period which includes but is not limited to antenatal care (ANC), delivery care postnatal care (PNC) services [1].

Each day, about 810 women died worldwide because of pregnancy and delivery-related complications in 2017. Out of this, 94% of the cases were contributed by developing countries while 86% is contributed by sub-Saharan Africa and South Asian countries. Sub-Saharan African countries contributed to nearly 66% of all world's maternal deaths [2]. The majority of the causes of maternal death are preventable or treatable if detected early. This fact is indicated by the huge gap in maternal mortality ratio between developed and developing countries and even between urban and rural women of the same country and in the same region [2, 3].

Globally there are inequalities in the utilization of maternal health services. This could hamper the sustainable development goal target to reduce maternal mortality through accessing equitable maternal health services. This inequality is mainly observed in Asian and African countries [4]. Research on the availability and accessibility of maternal health services across sub-Saharan African countries indicated that the majority of the countries full fill the minimum target criteria put by WHO at the national level. However, there is a huge gap in the accessibility and availability of maternal health services among different segments of the population with in same countries such as the Urban-rural gap, and rich-poor gap [5]. Studies in many low-income countries of Africa indicated that poor economic status, a far distance from health institutions, lack of transportation services to health facilities, poor quality of health services, poor family support, and cultural factors were identified as the most important barriers affecting maternal health service utilization [6].

In Ethiopia, direct obstetric causes i.e. complication directly related to pregnancy or childbirth accounts for nearly 75% of all maternal death cases while the other direct causes such as anemia (10.4%), ectopic pregnancy and indirect causes i.e. preexisting medical condition, malaria(3.55%), and HIV accounts the left 25 percent of maternal death cases. Hemorrhage which contributes to nearly 30% of maternal deaths is the leading of all direct causes. The majority of maternal deaths happened during the post-partum period and because of this focus should be given to this time [7–9].

As Ethiopian demographic and health survey (EDHS) of 2019 data revealed, there is an increment in the number of maternal health service coverage and utilization i.e. antenatal care, delivery service, and post-natal service utilization at national wide when compared to all

previously conducted surveys. About 74% of women received at least one antenatal care (ANC) and 43% of them received four and above ANC from a skilled health professional in the last five years preceding the survey. When compared with EDHS 2016, the percentage of mothers who contacted four ANC visits and above increased from 32% to 43%. However, the current ANC coverage is not equally shared among urban and rural and among different wealth quartiles. The current Ethiopian ANC coverage is below 80%, the minimum ANC coverage recommended by WHO. In the Oromia region, only 26% of the women who gave birth in the last 2 years received PNC from a health professional within 48 hours of post-partum [10].

With the concept of universal primary health service coverage, and to overcome the challenges of universal inaccessibility and low maternal health service utilization, the government of Ethiopia has gone through multi-dimensional approaches and strategies. One of them was the introduction of health extension programs in 2004 in rural areas of the country, fee exemption for maternal and child health-related services, and the recent introduction of ambulance services to rural sub-districts of the country [11–14].

Despite these, all efforts and strategies being applied by the government, and the partner organizations, maternal health service utilization is low in the general population and very low in rural communities of the West Shoa Zone specifically [10, 15–17].

Therefore, the barriers hindering rural women from utilizing the MCH service may not be identified only by quantitative study and needs detailed understanding from its base on socioeconomic and cultural aspects of the community level. Therefore, this study intended to identify and describe barriers contributing to low maternal health service utilization in selected rural districts of the west Shoa zone of Oromia regional state, Ethiopia.

## II. Methods

West Shoa Zone is one of the 20 Zones in the Oromia regional state and was founded in the west direction from Addis Ababa, the capital city of Ethiopia. It has 22 rural districts. There were about 494,213(18.63%) childbearing age women in the Zone. Out of the total reproductive-age women, about 92,000 (3.47%) of the women are expected to be pregnant each year. There were 8 hospitals, 91 health centers, and 514 health posts in the Zone. In general, the west Shoa Zone health center coverage (in terms of availability) was 92.6% and the hospital coverage was 30%. Three districts of the West Shoa Zone which includes: Dano, Elfata, and Midakegni were selected purposively for this study. This study was conducted in rural kebeles of the above-mentioned three districts. These districts were selected purposively based on the report of their last year's (2020) performance on maternal health service utilization obtained from the Zonal health office, where the ANC follow-up and Institutional delivery were the lowest among the Districts in the West Shoa Zone [18].

The study was conducted from February 01 to April 30/2021.

### a. Study design

A community-based qualitative phenomenological (focusing on the lived experience of women who gave birth at home) approach was used to explore the demand-supply side barriers affecting the utilization of maternal health services i.e. ANC follow-up and institutional delivery in rural communities of the three selected districts.

### b. Study population and sampling techniques

The study participants for the Key Informant Interview (KII) were leaders from selected districts i.e. head of the district health office, the catchment area health center director (PHCD), health extension workers of each selected kebeles (sub-districts), and the head of the MCH

department of each selected health center. The senior HEWs were selected for KII. Accordingly, 12 KII were conducted. Each of the key informant interviews was held at the actual working place of each participant. The study participants for KII were selected purposively based on their direct connection with maternal health services issues.

The study participants for the focal group discussion (FGD) were women who gave birth at home in the last 12 months before the data collection period. Accordingly, Six FGDs were conducted. Each focus group discussion had 8–10 members and took 45 to 60 minutes. The FGD was held at their respective catchment area health posts. To make the discussion environment conducive, coffee and tea refreshment (which is culturally preferable among the Ethiopian population) was arranged.

To get a diversified idea, the selection of the FGD participants considered the geographical location of the kebeles. Accordingly, participants were selected from the furthest, middle, and nearest kebeles. The study participants for IDI were women who gave birth at home in the last 12 months. Accordingly, 9 IDI were conducted. The IDIs were also conducted at the women's kebele health posts. The FGD and IDI participants were identified and included in the study through the facilitation of respective kebeles HEWs on the purposive selection method. The women who participated in IDI were not included in the FGD. The maximum number of the FGD and IDI were determined by the point at which there was no new idea become flourished from the participants i.e. redundancy of ideas.

### c. Operational definitions

**Maternal health services.**   In this study context the term maternal health service represents only antenatal care and institutional delivery.

### d. Data collection tools and methods

For both KII and IDI data collection, semi-structured, both open and close-ended questionnaires were developed in English and translated into Afan Oromo (S1 File). The discussion guide was prepared for FGD in the English language and translated into Afan Oromo. The background data of each study participant was collected using a questionnaire (self-administered and interviewer-administered). Data collection was conducted by trained data collectors who were fluent in Afan Oromo, and have an experience in qualitative data collection. KIIs, IDIs, and FGDs discussions were tape-recorded and field notes were also captured. The FGDs were facilitated by two data collectors and one supervisor. The data collector/facilitator raised the questions, balance the discussion among the discussants, asked questions for further clarification, tape-recorded the discussions, and took field notes.

### e. Data quality control

To ensure the quality of the collected data, a pre-test was conducted and the in-depth interview and FGD guide was modified based on the feedback obtained during the pre-test. In addition, data collectors/facilitators' recruitment was based on their prior experience in qualitative data collection, and also a one-day training was given for data collectors and facilitators. During data collection in the field, close supervision was held by the authors and there was also discussion on daily basis on challenges faced during the data collection and immediate solutions were put.

### f. Data processing and analysis

To enable the opportunity for triangulation of ideas, first KIIs were conducted followed by FGDs and IDIs. The audio-recorded data were transcribed verbatim on daily basis, and field

notes and Memos were arranged. Then, the data were translated into the English language. The translated data were transcribed to electronic format and imported to **MAXQDA-22 software** (Qualitative data analysis Software). Using the software, the imported data were first cleaned for their suitability for analysis and then coded (assigned category). Before coding started, a detailed re-reading of all transcription was done. After the detailed and iterative reading of the transcription, categories were developed by *the start list method* i.e. based on the researchers' experience, the topic of inquiry, and based on the existing literature. This method is generally the deductive approach. After the whole data (KIIs, IDIs, and FGDs) were coded and sub-coded, themes were generated. Finally, the analysis was done based on the generated themes. The data were analyzed by the members of the research themes who have an experience in qualitative data analysis using MAXQDA software.

## g. Ethics approval and consent to participate

The Ethical clearance was obtained from the Institutional ethical review Board (IRB) of the College of Medicine and Health Sciences, of Ambo University. The letters of support were written to the concerned body. During fieldwork, permission and interest to participate were requested after briefing the objective of the study. Both verbal informed and written consent were obtained from the study participants. Confidentiality of the data to be collected was secured and will not be transferred to any other third party. To ensure the confidentiality of the participants, the participants' names were not mentioned during interviews or FGD discussion, or report writing. Instead, the term Mr., Miss was used during KII and IDI, and code numbers were used during FGDs. This study is reported in accordance with the Consolidated criteria for reporting qualitative studies (COREQ) (S2 File).

## III. Results

A total of 6 FGD among women who gave birth at home in the last 12 months of the data collection period, 12 key informant interviews with the head of the district health office, Primary Health care director, maternal and child health coordinator, and health extension workers and 9 In-depth interviews with mothers who gave birth at home in the last 12 months were conducted.

The women who have participated in the study have different experiences with ANC follow-up and institutional delivery at different pregnancies. Some of the women have a history of ANC follow-up in the previous pregnancy, and some have a history of institutional delivery in previous delivery.

With the mean of 27 years, the age of the women who participated in FGD and IDI reaches from 18 to 38 years. Of the total 57 women who participated in the FGD and IDI, 31 of them didn't attend any formal education, 17 of them attended elementary level (5–8) and 9 of them attended primary education (1–4). All of the participants were from farmer families. With an average family size of three children, almost all of the study participants were multipara.

Following coding of the transcribed KII, IDI, and FGD data, and intensive re-reading of the transcribed documents, three themes were identified using the *start list method (the deductive method)*. The themes are individual-level barriers (demand side), socio-economic/geographic barriers, and health facility factors (provider side barriers) Table 1. The lists of identified barriers and how they interact are indicated in (S1 Fig).

All of the women who participated in the FGDs and IDI agreed and described the importance of attending ANC follow-up during pregnancy and giving birth at a health institution. However, because of the barriers, they have raised during the discussion and interview, they are unable to utilize the maternal health service specifically ANC and institutional delivery.

**Table 1. Identified themes on barriers to maternal health service utilization, Dano, Midakegn, and Ilfata Districts, West Shoa Zone, Oromia, 2021.**

| s. n | Identified themes | Categories |
|---|---|---|
| | Individual-level barriers | poor awareness, not knowing own EDD, forgetting appointment dates, precipitated labor, lack of BPCR, women's negligence, and shyness to give lab samples/being exposed to the opposite sex |
| | Health Facility-related barriers | Shortage of manpower, unavailability of Ambulance, un accessibility of ambulance, absence of regular pregnant women's conference, poor quality of service and welcoming at H/C, HEWs not accessible, poor advising |
| | Socio-Economic and Geographical Barriers | Distance from the health center, unavailability of transportation, unsuitable road, economic problem, fear of opportunistic costs, cultural preference, and adaption |

*EDD-estimated date of delivery *BPCR-Birth preparedness and complication readiness *HEWs- health extension workers *H/C-Health center

## Individual-level barriers affecting maternal health service utilization

These are barriers related to women own personal reasons or factors related to her pregnancy. These include poor awareness related to complications that can happen during pregnancy and childbirth, not knowing the estimated date of delivery (EDD), forgetting the appointment date, short laboring duration, lack of birth preparedness and complication readiness, negligence, being shyness and reluctance to give such as feces for lab test during ANC.

During the KII, it was identified that the women's awareness related to the number of ANC visits, danger signs during pregnancies, and complications related to home delivery were low. One of the KII participants said:

"…I am not confident enough to speak that all mothers of our woreda know the number of ANC she should attend during pregnancy. Some pregnant women from rural areas think that a single ANC visit is enough. As a result of this, women come to health centers for ANC at some point in time during their pregnancy or they may come for delivery without attending a single ANC. They don't think ANC follow-up is critical and must…"

(PHCD from Dano District)

Not knowing her EDD or forgetting her appointment date is also a barrier hindering women from using ANC follow-up and/or institutional delivery. If the woman doesn't know her estimated date of delivery, she will be less likely to prepare for delivery. The other commonly mentioned reason for giving birth at home among all FGD discussants and in-depth interview participants was the sudden onset of labor and short labor duration. The sudden onset of labor is mainly related to the pregnant mother's not knowing her EDD. There was also a misunderstanding among FGD discussants about HI delivery. That was unless there is a complication during labor, no need to go to HI. The issue of mentioning short labor duration is commonly raised among multiparas' women. One of the IDI participants said,

"…I gave birth to my first child at the health center. During that time the labor duration was elongated. I faced many complications, I vomit, and I became unconscious during labor. Because of that, my family took me to the health center and I give birth at the health center. But, on my second child, since I didn't face any problems, I gave birth at home. My labor was not prolonged. I started labor, and immediately I gave birth safely. …"

(IDI Participants from Ilfata District).

"... *Especially, multipara women don't want to deliver at HI. This could be due to two reasons: multipara women don't stay a long time with labor. And also if they previously gave birth at home, they will be reluctant to come to the health center for delivery. They say, since I gave my first or 2nd child at home, I didn't face any challenges during my previous child, why should I go to the HI. the second reason I think could be is Some of the women don't want to be seen by a person whom they don't know...*"

*(HEWs from Danno District).*

One of the FGD discussants described what has happened to her as a result of not knowing her EDD/gestational age and who helped her during labor as follows:

"*...Since I don't know my gestational age, when I first feel labor pain, I understood it as false labor (lafjalee). But it gets intense and finally, I gave birth at my home safely within short hours of labor onset. I was alone, nobody was with me. Later on, my husband joined me and helped me in cutting the newborn's umbilicus...*"

*(FGD discussants from Midekagni District)*

The other IDI participant from Ilfata explained the misunderstanding related to missing appointment date and its consequences saying:

"*... I forgot my appointment date during my ANC follow-up. I remembered my appointment date after the date was passed. Then I decided not to go to the health center. I felt the health center doesn't welcome me since the appointment date is already passed. I also gave birth at my home safely. I didn't come to this health center since I didn't attend my ANC as per health professionals' recommendation, I thought ANC follow up is prerequisite to give birth at health institution...*"

## Health facility-related barriers

These are barriers related to health facilities i.e. Health posts, health centers, and hospitals. The barriers identified as health facility-related barriers were: a shortage of manpower, un-availability/un accessibility of ambulances services, poor out-reach services at the community level (discontinuation of pregnant women's conference), Health professionals' poor welcoming and counseling approach during ANC visits, and delivery, and inaccessibility of HEWs at health posts. Fear of infections (especially HBV).

During our discussion with KII, FGD discussants, and IDI participants' manpower shortage was raised as a critical barrier for poor/low maternal health service (especially ANC and delivery service) utilization.

One of the KII participants explained the challenges they were facing in their health center as a result of manpower shortage as follows:

"*...The serious challenge hindering us from delivering MCH services is a shortage of manpower. You know, the majority of the services we provide here at the health center level are MCH-related services (ANC, delivery, Immunization.). We are two midwives in this health center. My coworker is not on duty today because she was on duty last night. So, I am delivering all services for MCH in our health center i.e. delivery, ANC, FP, and immunization being*"

*alone. As you know, attending a single delivery takes a long hour. While I am attending delivery at the delivery room, clients may seek FP or ANC service at MCH OPD. At that point, the client may stay some long hours to get service and such a long waiting time will disgust the client and may cause them not to come back again. . ."*

*(MCH head at H/C in Danno district)*

One of the FGD discussants from the same health center witnesses the above idea saying:

*". . .I visited this health center on three different days for ANC. During those three days, I was unable to access health professionals. On the fourth day, I went to the record room and the record room personnel told me as I can't access the service. Because no health professional will treat me. He refused to give me a patient card. Then I went back, and I didn't come back again. I gave birth at my home. . ."*

*(FGD discussants from Danno District)*

*". . .We do have only two midwives working in the MCH department. One of them was on duty last night and she is on rest today. The one on duty is working in the MCH department giving FP services, ANC follow-up services, delivery services, and post-natal care services alone. As you can observe now, there is a long waiting time. Almost all of these women are from rural areas. They will be upset with such a long waiting time and they may not come back again for the next appointment. I can't confidently say we are giving quality MCH care here. This is because of the imbalance between the number of health providers we have and the number of communities we are serving. . ."*

*(PHCD from Dano District).*

During our discussion with the study participants, it was also raised as unavailability and/or inaccessibility of the transportation service was causing women not to utilize ANC service and institutional delivery. Ambulance, the ideally expected transportation service during emergency/labor is also discussed by the participants as it is unavailable and inaccessible.

*". . .First I start to feel labor pain. While my husband is in search of transport, I gave birth. So, it is a transportation problem that hinders us to give birth at health institutions. . ."*

*(IDI participant at Ilfata District)*

*". . .Sometimes women give birth at home while she is waiting for an ambulance. Especially for the majority of the women who face labor during the nighttime, it is difficult to access transportation. When we call for an Ambulance we were told it is not available because of any reason. In such a case, the only existing option is giving birth at home. . ."*

*(FGD discussant at Midekagni)*

The far distance of the health facility is also another critical challenge presented as a factor hindering MCH utilization by study participants. This problem is more critical in areas where transportation service is not available, and pregnant women are obliged to travel on foot to get services.

*". . .As you have observed, the health center is too far from our kebele, we are expected to travel more than two hours to receive ANC and delivery services. This is something difficult*

*for a pregnant mother to travel such a long distance on foot. Even we can't get transportation. So, we should have to travel on foot for more than two hours. Including me, this is hindering many of our kebele's pregnant women to utilize the maternal and child health services as expected. So, I wish we can access basic services here in our nearby health posts. . ."*

*(IDI participant from Midekegni)*

*". . . As I have told you earlier, nine of our woreda's kebeles are remote. It takes 6 to 9 hours to reach the health center. Public transportation is not available at the needed time. This problem is the worst, and even there are kebeles where there is no road and it is hard for the ambulance car to pick the laboring mothers. . ."*

*(Midekegni Woreda Health office).*

One of the KII who participated in the study described the challenges related to the inaccessibility of the ambulance as follows:

*". . .Mothers from far kebeles are also not accessing Ambulance service during laboring. The inaccessibility of Ambulance is because for many reasons. One is lack of driver. We do have four Ambulances. However, since the structure (the rule) allows a maximum of 2 drivers per Woreda, two of the Ambulances are not giving service. As a result of this Ambulances are controlled centrally by woreda health offices, and the health center will call on the phone from Woreda when they need the Ambulance. This will elongate the chain of services and may result in a service delay. . ."*

*". . .Most of the time, the laboring women cannot access ambulance services because of the unavailability of the car. When we call the ambulance from the woreda health office, they sometimes say the car was sent to the hospital for referral, or they may say the car is not functional and it is in the garage. . ."*

*(FGD participants from Dano District).*

Distance from health institutions affects MCH service utilization in different ways. One is women from economically poor families may not get transportation costs. The other one is if the women come to give birth from such a long distance, it may take a minimum of three days to come back home (one day for coming to H/C, one day for delivering, and one day for going back to her home). During these all days, there are many costs like food, transportation, and other opportunistic costs like leaving their children and cattle and plowing lands alone. Because of fear this all costs, the mother prefers to give birth at home.

One of the health professionals who participated in the study described this scenario as follows:

*". . .Ambulance can't reach nearly nine kebeles of our woreda. Look, the mother will come traveling such a long road. Some of the mothers die on the road, and when they reach the health center, the case may be beyond our capacity. Then, we will be obliged to refer to Gedo Hospital. Imagine the duration and complication the mothers are facing starting from home till she reaches hospital. . ."*

*(PHCD from Midakegni District)*

The other provider-side barrier hindering mothers from using MCH service is the home-to-a-home visit by HEWs is too weak. HEWs are not visiting pregnant women's homes and are not advising on the importance of ANC, risks, and complications related to home delivery.

*". . .As I have told you, I am working at this Health Post alone. The workload is too high, and I can't cover the whole need of the people. It is expected of me to conduct a home visit and deliver immunization and FP onsite, and I am also a member of the kebele cabinet. If I went for training somewhere, the HP will be closed for days or weeks. This could be another source of complaint and dissatisfaction for pregnant women who may miss me during my absence. . ."*

*(HEW from Dano District).*

*". . . The HEWs are not covering the home visit as per standard. Because of the workload on HEWs and the lack of sufficient HEWs at the health post level, they fail to cover what is expected from them. As you know, the rural population is sparsely populated, and it is very difficult for one or two of the HEWs to cover all HHs. In addition to their regular responsibilities, HEWs are working as a member of the kebele cabinet, and are engaged in political activities like revenue collection. These will overwhelm the health extension workers and make them not achieve the goals they are assigned for. Health posts are open only once or twice per week just only for giving regular vaccines. . ."*

*(PHC director at Danno District).*

*". . .some of the HEWs quit their job, some are on maternity leave and some are on long-term training (more than 2 years) at colleges/universities. Since the population of rural areas is sparsely distributed, it is very difficult for only one or two HEW/S to address the whole HHs in the kebele as needed. Therefore, because of the shortage of HEWs, health posts are closed most of the time, and pregnant women are not accessing the intended services at health posts. . ."*

*(PHC director at Ilfata District)*

*". . .And most of the time health extension workers in our kebele are also not available at working time. There is a time when we go to the health post and we are unable to get the service. They live in a town which is far from the health posts. The HEWs come to health posts a maximum of one or two days per week. So, I wish if these things are also improved. . ."*

*(FGD discussants from Midakegni District)*

The other barriers to poor MCH utilization among pregnant women identified in this study are poor welcoming approach from health providers during ANC visits and delivery and poor cleanness of the delivery room.

*". . .I gave birth once at the health center. The approach was not good there. The care provider insulted me. They don't care for laboring mothers. Especially Female Health professionals are not welcoming us. Male Health professionals are relatively compassionate (emphatic). . ."*

*(FGD discussant from Midekegni)*

*". . .For instance, in my kebele, women prefer some of the HEWs over the others. They don't want to be seen by some HEWs. They escape. This is due to the skills and approach HEWs have. Unless you approach the mother with an attractive approach like your family, welcoming her with a good approach, emphatically listening to her, she doesn't accept what you advise. . ."*

*(HEWs from Midekagni)*

*". . .sometimes I accompany laboring mother to H/C, There, delivery rooms were not clean and attractive. You know, women from rural areas are very sensitive, shy and even don't speak what they need. There is a time when a rubber sheet on the laboring coach is not clean and covered with the blood of other women. The laboring women are ordered to sleep over such an uncleaned bed without even making a bed with the women's sheet. Such women are less likely to come back to health centers again in the future. . ."*

*(HEWs from Dano District)*

In addition, one of the FGD discussants portrayed what she has faced as follows:

*". . . I gave birth to my first child here at this health center. Immediately after I gave birth, all the health care providers left me alone and went to sleep. That night, I faced bleeding at the health center. No one helped me there. . .. One week later I came back to the same health center because my child gets sick. Even we were unable to get transportation. After many challenges, we reached the health center. While I am on the health center campus, my child died without getting any health service. In the next pregnancy, I lost my trust and I didn't come to the health center even for ANC. I gave birth at my home. If I don't get service here, why do I come?"*

*(25 years old woman FGD discussants Dano District)*

The other IDI participant also witnessed the problem related to poor welcoming from the providers' side saying:

*". . .When I went to give birth to my first child, my birth attendant was a female midwife. I felt serious labor pain and I shouted. She, my birth attendant mocked me saying, how did you get pregnant first? Later on, she went to sleep and she didn't visit me over the night. It was an unexpected event I encounter during that night. . ."*

*(IDI participants from Midakegni District)*

## IV. Discussion

In this study, different barriers hindering mothers from utilizing maternal health care specifically, ANC and institutional delivery were identified. From the demand side, women's Poor awareness related to complications that can happen during pregnancy and childbirth, not knowing the estimated date of delivery (EDD), forgetting the appointment date, short laboring duration, lack of birth preparedness, and complication readiness, misinformation about ID were identified.

Having awareness and sufficient information about maternal health care such as the number of ANC, where and when to utilize it, danger signs and complications during pregnancy and childbirth, birth plan, and estimated date of delivery are the pre-conditions that the women should have to know so that she can utilize the service. This study revealed that there is a gap in the above-mentioned issues. One of the major reasons why the women were giving birth at home was not knowing their estimated date of delivery (EDD). If the woman doesn't know her EDD, she will be less likely to prepare for birth i.e. deciding on birthplace and preparing for transportation and other matters. So, as a result of not knowing her EDD, she may face sudden labor onset and she will give birth at home. One of the underlying reasons for not knowing EDD is not attending ANC. Once the women attend ANC, she will be more likely to be told about her EDD during ANC [19, 20].

In addition, short labor duration (precipitated labor) and not facing complications during labor were also other major reasons contributing to home delivery. These two things are miss-information we identified in this study. Rural women are considering giving birth at health institutions only if the labor is prolonged and the women faced complications. This miss information is common among women who have a history of home delivery in addition to the current birth. This finding is similar to the finding of a study conducted in Eastern Oromia, in the Amhara region, and in Sidama where the women reported not going to the health institution for ANC and childbirth unless they get sick or felt pain during pregnancy [20–22].

Women's forgetting their next time appointment date is also identified as a barrier affecting utilizing ANC and institutional delivery. If the woman remembered her next appointment date after the date passed she consider it to be already past and did not need to go to HI again. This indicates that there is a gap in women's knowledge of the ANC timing.

From supply-side barriers, a shortage of manpower, un-availability/un accessibility of ambulances services, poor out-reach services at the community level (discontinuation of pregnant women's conference), Health professionals' poor welcoming and counseling approach during ANC visits, and delivery, and inaccessibility of HEWs at health posts, Fear of infections (especially Hepatitis B Virus) were identified.

One of the building blocks of the health system is manpower. Shortage of manpower has a huge impact on the quality of the service and finally may end up with poor service. If the pregnant woman and/ or laboring woman are not accessing the quality health service they need once upon a time at a certain health facility, they will lose trust in the health system and may not utilize the service again. One of the challenges related to the shortage of manpower is the long waiting time at health institutions, and not accessing all services they need at a single point in the health facility. As a result of this, the women may be appointed for another time or referred to another health facility because of no health professional. Rural women are coming from far distances penetrating multiple challenges such as socio-economic and/or transportation challenges. In areas where there is a shortage of manpower, let alone community-based awareness creation such as pregnant women's conferences, even health facility-based routine services will be of poor quality [23].

Even though it is expected from the HEWs to screen pregnant women and send them to nearby health centers, and facilitate the "**pregnant women's conference**" in their catchment kebeles (sub districts), the HEWs were not doing such activities. This is because of two main reasons. One is there is a shortage of the number of the HEW compared to the population they were serving. Since some of the HEWs terminated their job and some of them were on long-term training. The second is that HEWs are forced to engage in extra duties such as political activities in addition to the health extension packages (HEP). HEWs also live far away from their health posts where it takes more than 2 hours to travel on foot. In such cases, they are not accessible at the health post level at the needed time. These all challenges are overwhelming the HEWs and hindering them not to delivering the ideally expected services and reaching pregnant women. This finding corroborates the study conducted in Abuna Gindabarat of the Oromia region where Women's ANC service utilization from HEWs was very low [17].

Distance from health centers is yet a great challenge for rural women to access maternal health services. Though there is an improvement over time in health service coverage, rural women are facing challenges to get maternal health services because of distances and the geographical unsuitability of roads for transportation. Many women from rural areas are facing transportation challenges. The transportation problem is either there is no transportation at the needed time or the women may have no money for transportation during ANC or laboring. Ideally, it is expected that all sub-districts (kebeles) should have to access an Ambulance service for laboring mothers. However, this is not true in many of the sub-districts of the study

areas. Pregnant women are expected to travel on foot for more than six hours to days to access maternal health services. Inaccessibility of public transportation and lack of transportation costs make the challenges the worst. Similar findings were reported in other parts of the country and other African countries like Ghana [24]. Laboring women were not accessing Ambulance services during emergency times. This is due to many reasons. One is the number of available ambulance cars doesn't match the population being served. There were only one or two ambulance cars per district. That single ambulance will serve to bring laboring mothers from the community to the health center and from the health center to referral hospitals. During such time, there will be a need overlap and some of the women may not access the ambulance service immediately at the needed time. Sometimes, the ambulance car faces a technical problem and may stay more than a few months to be repaired. These also contribute to home delivery too [25, 26]. In addition, a woman who gave birth at HI needs transportation to come back home. But, it is a challenge to access transportation. They were requested to pay for the ambulance fuel. Not all mothers can afford that. In fear of this, some mothers again prefer home delivery [22].

The other critical supply-side barriers hindering women not to using ANC and institutional delivery were poor respectful care during ANC, and labor and childbirth. Laboring mothers are in very severe pain. They need both medical and psychological support. Any negative reaction from health professionals will demoralize them and hinder them not to coming for maternal health services in the future. Mocking at them while they are in labor pain or when they request help, if the labor room is not clean and attractive, if the health professional doesn't approach mothers who seek ANC or delivery services with empathy and respect, they will never trust the health professional again and they will be hesitant to utilize the service another time [22, 24, 27].

Infectious diseases transmitted by contact with body fluid were also contributing to home delivery. HBV is one such disease. In the area where the HBV prevalence is relatively high among pregnant women, health professionals become hesitant to attend the labor, since they believe it is risky for them to encounter the infection. In such cases, the health professional prefers to refer the laboring woman to the hospitals where the HBV vaccine for health professionals can be accessible. Since they can't afford different costs almost all of the women who will be referred to the hospital will go back and give birth at home. This problem is challenging in the district where the HBV prevalence is high and the HBV vaccine is not available. This finding is unique to the Midekegni district of the West Shoa Zone.

## V. Limitations of the study

We have thought a lot to get feedback from participants on the final compiled finding report. However, since almost all of the IDI and FGD participants couldn't read or write, it was difficult to us to request the participants' feedback on our final report.

## VI. Conclusion

In this study, we have tried to explore barriers hindering women from using ANC and Institutional delivery in the rural area of the West Shoa Zone. Our findings revealed that pregnant women strongly agree on the importance and advantage of maternal health service utilization, and they have the interest and need to utilize ANC and delivery care at health facilities. However, many barriers such as low awareness on when and where to use the services, not knowing scheduling of ANC, misinformation about ANC and institutional delivery (perception of seeking ANC or ID only if there are complications), not knowing their EDD, shortage of manpower at health post and health center level which is leading to poor quality health service

(especially poor respectful care), unavailability of HEWs at the needed time, far distance health facilities, and unavailability or un accessibility of ambulance services during an emergency time, lack of public transportation services and fear of costs related to ID were hindering the women from utilizing ANC and ID.

In general, rural women are facing many challenges yet to accessing and utilizing maternal health services. When we frame these barriers according to Thaddeus and Maine's three delays model; all three delays are contributing to low ANC and ID service utilization. The majority of the barriers are those already identified more than ten years ago and becoming obstacles yet.

Ethiopia is working towards the achievement of the SDG of reducing MM to less than 140 deaths /100,000 live births, and reducing IMR to less than 12 deaths /1000 live births by 2030. To achieve these SDG targets, and to ensure Universal health service coverage, addressing barriers prohibiting a woman from using MCH care is of point to be targeted.

Since the demand-supply barriers to maternal health service utilization are interrelated, addressing only some of the barriers may not mean women can utilize the services. For instance, improving only women's awareness of MCH may not solve the problem if there is no physically available health institution. Therefore, addressing these barriers needs a comprehensive, multi-directional approach that can address both demand and supply barriers.

Therefore, we would like to forward the barrier-specific actions to be taken as follows:

## Action to be done to address demand-side barriers

To address the gap in the awareness of ANC scheduling, and misinformation about when and where to seek services, resuming and strengthening the regular "*pregnant woman's conference* (PWC)" will have a crucial contribution. Strengthening the community-based pregnant women's conference will have two advantages. One, it will help in addressing and rectifying the information gap and it will also contribute to demand creation on ANC and institutional delivery. To overcome challenges related to the shortage of manpower, task shifting can be taken as an alternative. Establishing a health development army (HAD), training them, and shifting some tasks of HEWs toward HDA can solve the manpower shortage at the community level.

## Action to be done to address supply-side barriers

Organizing regular outreach programs to those women from far-reaching sub-districts could alleviate the transportation shortage. In addition, organizing frequent training on Compassionate, and respectful care (CRC) for health professionals is needed. Overall, addressing the structural issue such as the physical unavailability of health institutions may need a long-term plan from the government office.

## Supporting information

**S1 File. This is data collection tools for FGD, IDI and KII.**
(DOCX)

**S2 File. This is consolidated criteria for reporting qualitative research.**
(DOC)

**S1 Fig. This is Fig 1 indicating interaction among different barriers.**
(PDF)

## Acknowledgments

We would like to thank Ambo University for covering the data collectors' costs and supervisors per diem. In addition, we would like to forward our heartfelt gratitude to our study participants, our data collectors, and Midakegni, Elfata, and Dano Health offices.

## Author Contributions

**Conceptualization:** Seifadin Ahmed Shallo, Deresa Bekele Daba, Abuzumeran Abubekar.

**Data curation:** Seifadin Ahmed Shallo, Abuzumeran Abubekar.

**Formal analysis:** Seifadin Ahmed Shallo, Deresa Bekele Daba.

**Investigation:** Seifadin Ahmed Shallo.

**Methodology:** Seifadin Ahmed Shallo.

**Resources:** Seifadin Ahmed Shallo.

**Software:** Seifadin Ahmed Shallo, Deresa Bekele Daba.

**Supervision:** Seifadin Ahmed Shallo, Deresa Bekele Daba, Abuzumeran Abubekar.

**Validation:** Seifadin Ahmed Shallo.

**Visualization:** Seifadin Ahmed Shallo.

**Writing – original draft:** Seifadin Ahmed Shallo.

**Writing – review & editing:** Seifadin Ahmed Shallo, Deresa Bekele Daba, Abuzumeran Abubekar.

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
