## [Decision Letter · Decision Letter 0]

4 Aug 2022

PONE-D-22-18718Title: Demand –supply-side barriers affecting maternal health service utilization among rural women of West Shoa Zone: A qualitative studyPLOS ONE

Dear Dr. Shallo,

Thank you for submitting your manuscript to PLOS ONE. After careful consideration, we feel that it has merit but does not fully meet PLOS ONE’s publication criteria as it currently stands. Therefore, we invite you to submit a revised version of the manuscript that addresses the points raised during the review process.

We look forward to receiving your revised manuscript.

Kind regards,

Dylan A Mordaunt, MD, MPH, FRACP

Academic Editor

PLOS ONE

Journal Requirements:

2. Please upload a copy of Figure 1, to which you refer in your text on page 7. If the figure is no longer to be included as part of the submission please remove all reference to it within the text.

Additional Editor Comments:

Thank you for your submission. The main feedback was for additional detail with overall minor suggestions.

With regards to the criteria for publication:

1. The study appears to present the results of original research.

2. Results reported do not appear to have been published elsewhere.

3. Experiments, statistics, and other analyses require further detail as described by the reviewers. I would also suggest that given the general readership of this journal that the epistemology and methodology are outlined in a bit more detail.

4. Conclusions are presented in an appropriate fashion and are supported by the data.

5. The article is presented in an intelligible fashion and is written in standard English.

6. The research meets all applicable standards for the ethics of experimentation and research integrity.

7. I would suggest the authors use a guideline or checklist for qualitative research such as the SRQR (https://www.equator-network.org/reporting-guidelines/srqr/) or COREQ (https://www.equator-network.org/reporting-guidelines/coreq/).

Reviewers' comments:

Reviewer's Responses to Questions

**Comments to the Author**

1. Is the manuscript technically sound, and do the data support the conclusions?

Reviewer #1: Yes

Reviewer #2: Yes

2. Has the statistical analysis been performed appropriately and rigorously? 

Reviewer #1: N/A

Reviewer #2: N/A

3. Have the authors made all data underlying the findings in their manuscript fully available?

Reviewer #1: Yes

Reviewer #2: Yes

4. Is the manuscript presented in an intelligible fashion and written in standard English?

Reviewer #1: No

Reviewer #2: Yes

5. Review Comments to the Author

Reviewer #1: Very interesting work on demand and supply side barriers. Please see the comments inserted in the paper. Please add some more global references to demand and supply side barrier analysis in the earlier section. The conclusion section needs to be strengthened. suggest the authors make some recommendations on easily addressable barriers and those that require more long term systemic changes

Reviewer #2: This manuscript explored an important topic of public health importance. Overall, the manuscript is well written. The authors made reasonable arguments why exploring demand-side barriers for maternal health care seeking is important in Ethiopia. Below are comments which may improve the manuscript;

1. Please describe how the authors decide upon the sample sizes for interviews and what sampling procedure was used.

2. Did the authors achieve data saturation? If yes, which type of data saturation was achieved?

3. Did the authors pretest the interview guidelines and made any modifications? If yes, how did they incorporate the feedback?

4. How did the authors ensure data quality?

5. Please include the reflexivity of the researchers included in data collection, analysis, and writing.

6. PLOS authors have the option to publish the peer review history of their article (what does this mean?). If published, this will include your full peer review and any attached files.

Reviewer #1: No

Reviewer #2: **Yes: **Nazia Binte Ali

---

## [Author Response · Author response to Decision Letter 0]

13 Aug 2022

Responses to reviewers/academic editor’s comments 

First of all we appreciate and would like to thank the academic Editor and reviewers for their constructive comments which we believe will strengthen the quality of our work. With this, we have addressed the comments and suggestions forwarded from reviewers and editorial offices as follows. 

Responses to Comments raised from academic editor: 

Response: we have already addressed the points as per the PLOS One template guide at first submission of the manuscript. We have also named the file accordingly 

2. Please upload a copy of Figure 1, to which you refer in your text on page 7. If the figure is no longer to be included as part of the submission please remove all reference to it within the text.

Response: the Figure is included (already there) on page 16 of the manuscript. Now, we have put as separate document and uploaded it with the revised manuscript. 

3. I would suggest the authors use a guideline or checklist for qualitative research such as the SRQR 

Responses: we have included the COREQ checklist as supplementary document and uploaded it. 

Responses to the Comments raised from Reviewers

Reviewer #1: Very interesting work on demand and supply side barriers. Please see the comments inserted in the paper. Please add some more global references to demand and supply side barrier analysis in the earlier section. The conclusion section needs to be strengthened. Suggest the authors make some recommendations on easily addressable barriers and those that require more long term systemic changes. 

Response: we have tried to add the global status of demand-supply barriers to maternal health service utilization in introduction part and indicated with track changes. We have also addressed the recommendations in terms of short term and long term intervention as per recommended. 

Reviewer #2: This manuscript explored an important topic of public health importance. Overall, the manuscript is well written. The authors made reasonable arguments why exploring demand-side barriers for maternal health care seeking is important in Ethiopia. Below are comments which may improve the manuscript;

1. Please describe how the authors decide upon the sample sizes for interviews and what sampling procedure was used.

Response: as we have tried to explain in the method sections of the manuscript, the key informant interview participants for the study were selected purposively based on the direct connection (coordinating the MCH service, giving leadership, delivering the service) they have with the maternal health service issues. (Mentioned under the study population and sampling techniques subsection of part of the method section). There are 4 government structures which are in charge of maternal health issue at district level i.e. District health office, PHC director, MCH head at each health center and HEWs at community level. Since we have conducted our study in three districts, we finally decided to conduct KII with 12 personnel i.e. four each districts. (3 district *4 structures=12 participants). 

In both KII and IDI, we have used purposive sampling technique. ((Mentioned under the study population and sampling techniques subsection of part of the method section).

2. Did the authors achieve data saturation? If yes, which type of data saturation was achieved?

Response: yes. The type of data saturation the authors used for FGD and IDI was the “redundancy of idea”. We have conducted both FGD and IDI to the level where no new idea comes out of the participants (the idea is repeated). 

3. Did the authors pretest the interview guidelines and made any modifications? If yes, how did they incorporate the feedback?

Response: thank you for this question. Actually e have pre tested our tools ahead of data collection. We have adjusted a lot. E.g. order of probing questions, time it take to complete, we have avoided some unrelated questions, and we also able to add some questions based on feed bac from the pre-test participants. Why we didn’t include in the manuscript is because of there is no subheading which invite us to include issue about data quality control as per the PLOS one journal manuscript preparation guideline. Now, we have included new subheadings (data quality control subtopic) 

4. How did the authors ensure data quality?

Response: yes. See the “data quality control” sub topics. (Newly added sub topics). 

5. Please include the reflexivity of the researchers included in data collection, analysis, and writing.

Response: we have included in the main document

---

## [Editor Report · Decision Letter 1]

22 Aug 2022

Title: Demand –supply-side barriers affecting maternal health service utilization among rural women of West Shoa Zone, Oromia, Ethiopia: A qualitative study

PONE-D-22-18718R1

Dear Dr. Shallo,

We’re pleased to inform you that your manuscript has been judged scientifically suitable for publication and will be formally accepted for publication once it meets all outstanding technical requirements.

Kind regards,

Dylan A Mordaunt, MD, MPH, FRACP

Academic Editor

PLOS ONE

Additional Editor Comments (optional):

Thank you for your resubmission. This now meets the criteria for publication.
---

## [Editor Report · Acceptance letter]

25 Aug 2022

PONE-D-22-18718R1 

Title: Demand –supply-side barriers affecting maternal health service utilization among rural women of West Shoa Zone, Oromia, Ethiopia: A qualitative study 

Dear Dr. Shallo:

I'm pleased to inform you that your manuscript has been deemed suitable for publication in PLOS ONE. Congratulations! Your manuscript is now with our production department. 

Kind regards, 

on behalf of

Associate Professor Dylan A Mordaunt 

Academic Editor

PLOS ONE